# communications
## earth & environment

# Microbial growth in actual martian regolith in the form of Mars meteorite EETA79001

Neveda Naz[1], Bijan F. Harandi[1,2], Jacob Newmark [1,2] & Samuel P. Kounaves [1✉]

Studies to understand the growth of organisms on Mars are hampered by the use of simulants to duplicate martian mineralogy and chemistry. Even though such materials are improving, no terrestrial simulant can replace a real martian sample. Here we report the use of actual martian regolith, in the form of Mars meteorite EETA79001 sawdust, to demonstrate its ability to support the growth of four microorganisms, *E. coli. Eucapsis* sp., *Chr20-20201027-1*, and *P. halocryophilus*, for up to 23 days under terrestrial conditions using regolith:water ratios from 4:1 to 1:10. If the EETA79001 sawdust is widely representative of regolith on the martian surface, our results imply that microbial life under appropriate conditions could have been present on Mars in the past and/or today in the subsurface, and that the regolith does not contain any bactericidal agents. The results of our study have implications not only for putative martian microbial life but also for building bio-sustainable human habitats on Mars.

[1] Department of Chemistry, Tufts University, Medford, MA 02155, USA. [2] These authors contributed equally: Bijan F. Harandi, Jacob Newmark.
✉email: samuel.kounaves@tufts.edu

A variety of landed and orbital missions to Mars over the past decades have revealed a hyperarid environment inimical to life as we know it. In addition to the constant bombardment by galactic cosmic rays, solar particles, and ultraviolet (UV) radiation[1,2], the surface material globally contains highly oxidizing oxychlorine compounds and radicals[3–5]. While the extreme physical and chemical environment on Mars has made any life on the surface unlikely, microbial life on Earth has increasingly been found to have adapted and thrived in a variety of extreme environments that decades ago would have been considered uninhabitable. This ability has been displayed by a variety of bacteria, fungi, and archaea, that are referred to as extremophiles, including species that have been shown to grow and survive under conditions similar to those that exist today on Mars[6–8]. It has been suggested that Mars could have had a habitable environment in its early history, including a thick $CO_2$ atmosphere and large bodies of liquid water[9,10]. During this time, and perhaps today in the subsurface or unique microniches, the accepted criteria for habitability may be met and contain the essential bio-nutrients, liquid water, and energy sources to support organisms similar to terrestrial extremophiles[11]. Understanding the interactions of microbial life with martian regolith will provide insight into potential environments on Mars that are or were habitable, and thus allow for better-focused searches and instrumentation on current and future Mars missions. (Note: The term regolith is used for both material constituted of primary rock particles which are not weathered, and also for material that has been exposed to the atmosphere and hydrolytically weathered and often referred to as soil[12]).

Lacking martian regolith to study, a wide variety of research has been focused on the use of regolith simulants in attempts to understand the chemistry and habitability of real martian regolith. Although no natural or synthetic simulant can represent the true surface composition of Mars, they have been studied in great depth. Characterization of the mineralogical and physicochemical properties of simulants, combined with data from Mars missions, has provided a better understanding of the martian surface regolith[13–15]. Many studies have reported on the response of bacteria with a variety of martian or terrestrial simulants under different environmental conditions, mineralogy, geochemistry, temperature, pressure, and radiation[16–24]. The conclusion of these studies has been that the martian simulants, and by inference martian regolith, have the capacity to potentially support some forms of life.

Two observations of microbial growth, within a whole Mars meteorite, have been reported. An earlier study described the microbial activity (probably belonging to a member of the *Actinomycetales*) within the ALH84001 martian meteorite from the Antarctic blue ice fields, likely introduced during its residency in the ice[25], and a more recent one that examined the in-situ interactions of microorganisms in microbially-assisted chemolithoautotrophic biotransformation within iron-rich material in the NWA7034 martian meteorite[26]. The authors showed by observing bioalterations at the nanometer scale, that the bacterium *Metallosphaera sedula* had a greater affinity for colonizing the meteorite compared to terrestrial rocks, potentially serving as a distinctive biological signature[26]. Whilst this type of data informs us on the chemolithotrophic mechanisms of certain bacteria, it provides limited information on the importance of whether martian regolith could be directly utilized as a potential growth substrate at the microbial level. Our work was aimed at directly observing the effect of martian regolith on the growth and continued survival of bacteria, where the sole nutrient source was the regolith.

**The EETA79001 Mars Meteorite.** Of the hundreds of meteorites and fragments identified as originating from Mars, most are either of insufficient mass to allow for the amount of material needed for bacterial growth experiments, and/or were not collected and curated under controlled conditions. Even more importantly, only a dozen meteorites have been definitively shown through noble-gas isotopic analyses to have originated from Mars[27]. One of the larger martian meteorites, designated as EETA79001, was recovered by the NASA/NSF ANSMET program in Antarctica in 1979 and weighed ~8 kg[28]. Although it has been extensively studied, it has never been used for directly investigating the growth of microbial organisms. Over the past several decades, as the sample was cut into smaller sections for distribution, a substantial amount of sawdust was generated and collected. The meteorite was cut at the NASA-Johnson Space Center in a nitrogen-filled cabinet using a diamond saw without the use of any oil or water. Studies have shown contamination during the sawing process to have been minimal and are well understood[29]. This is the first time, to the best of our knowledge, that martian regolith in the form of the EETA79001 sawdust has been directly used as a substrate to test the hypothesis that it contains the nutrients and minerals necessary to support bacterial growth and that bactericidal agents are absent or their effects minimal.

The appropriateness of the EETA79001 sawdust as a representative sample of the present martian regolith is supported by its young age, having been dated at $\sim 1.7 \times 10^8$ years, with an ejection age of $\sim 6.5 \times 10^5$ years[30]. More importantly, is its similarity with recent in-situ analyses of martian regolith. The most relevant of these, in terms of aqueous leaching of the regolith's soluble components, was that provided by the Phoenix Mars Lander's Wet Chemistry Laboratory (WCL) in 2008 at its landing site in the northern plains of Mars. The WCL performed the only wet chemical analyses to date of the ionic composition (soluble inorganic salts) of the martian regolith. The 1:25 regolith:water mixture was found to be slightly alkaline (pH 7.7), with low salinity (~1400 $\mu S\,cm^{-1}$), and containing a variety of ion species that included potassium ($K^+$), sodium ($Na^+$), magnesium ($Mg^{2+}$), calcium ($Ca^{2+}$) sulfate ($SO_4^{2-}$), and chloride ($Cl^-$)[31,32]. Overall, the aqueous geochemistry of the martian regolith at the landing site was not substantially different from terrestrial regolith from hyperarid environments such as the Atacama Desert or the Dry Valleys of Antarctica[33]. One major difference was the relatively high concentration of perchlorate ($ClO_4^-$) at the Phoenix landing site[31]. Our previous work has shown that the concentrations (Supplementary Table S1) of most ions and conductivity of the EETA79001 sample are only 2–8% of the Mars regolith samples at the Phoenix site, except for $Ca^{2+}$ (~20%), $K^+$ (~0.5%), and $ClO_4^-$ (~0.04%)[33]. Even though the concentrations are lower in the EETA79001, the overall ratios of the ions are very similar. With the exception of $ClO_4^-$, both the EETA79001 and WCL samples are dominated by $SO_4^{2-}$, $NO_3^-$, $Na^+$, $Mg^{2+}$, and $Ca^{2+}$. This similarity provides confidence that the meteorite-derived regolith is representative of martian surface regolith. The EETA79001 leachate includes sufficient amounts of inorganic nutrients necessary for bacterial growth, including $PO_4^{3-}$, $NO_3^-$, and $SO_4^{2-}$, and especially $Ca^{2+}$ and $Mg^{2+}$, ions necessary for growth and cell maintenance[34,35].

The martian landing sites to date have been shown to contain an average of a 0.6 wt% of $ClO_4^-$[4,31,36], with concentrations detected by the Sample Analysis at Mars (SAM) instrument package on board the Curiosity rover varying between 0.07–1.3 wt%[37]. Although the concentration of $ClO_4^-$ in the EETA79001 (~600 ppb)[30] is much less than has been measured on Mars, it is well above the typical levels found on Earth[38]. This lower $ClO_4^-$ concentration in EETA79001 may be due to its age (~180 Myr), its ejection site where less $ClO_4^-$ may have been

present, or the radiolysis of the $ClO_4^-$ during its travel to Earth[30,39]. Although toxic to some microorganisms, numerous studies have shown that $ClO_4^-$ can be used by many bacteria as a terminal electron acceptor for anaerobic respiration, especially by halophilic extremophiles, thus fueling growth[40–42]. Given the concentration of $ClO_4^-$ present in the EETA79001, it was not expected to have any effect on any of the bacteria used[43–45]. The presence of $ClO_4^-$ on Mars could be a benefit for microbial life, allowing growth at much lower temperatures since $ClO_4^-$ salts have the ability to lower the freezing point of water, allowing it to exist as a liquid at temperatures down to −70 °C, thus providing subsurface pockets of saline water for life[43,46,47].

In this study, we used the EETA79001 Mars meteorite sawdust to directly address the question of whether martian regolith in appropriate aqueous environments on Mars, past or present, could support microbial life. Our results imply that microbial life under appropriate conditions could have been present on Mars in the past and/or today in the subsurface, and that the regolith does not contain any bactericidal agents. The results of our study have implications not only for putative martian microbial life but also for building bio-sustainable human habitats on Mars based on cyanobacteria and martian regolith.

## Results and discussion

Using the EETA79001 sawdust as a substrate with different ratios of regolith:water (Fig. 1), we have investigated the growth and survival of four non-spore-forming microorganisms under terrestrial conditions. The microorganisms were selected to compare two extremophiles, a cyanobacterium isolated from the Atacama Desert Chr20-20201027-1 (*Chr20*) and the bacterium *Planococcus halocryophilus* (*P. halocryophilus*), with two widespread and commonly used microorganisms, the non-filamentous cyanobacterium *Eucapsis* sp. (*Eucapsis*) and the bacterium *Escherichia coli* B (*E. coli*). The bacteria were chosen so as to better understand the role of water and the minimal mineral nutrients in the regolith matrix necessary for growth and survival. Further, the choices serve to compare two typical bacteria with two extremophiles to characterize how these species are suited for growth in martian regolith[43,48,49]. The rationale and characterization of the microorganisms selected is further described in Supplementary Note 1.

**Cyanobacteria: *Eucapsis* and *Chr20*.** The growth and survival of the two cyanobacteria, *Eucapsis* and *Chr20*, in EETA79001 regolith under terrestrial conditions, were evaluated for 22 days at regolith:water ratios of 2:1, 1:2, 1:5, and 1:10. Additionally, the 4:1 ratio was used as a dry growth control with only the cyanobacteria inoculum solution added to the regolith, and a 0:1 ratio was used as a water growth control containing no regolith. A complete description of the culturing, growth, enumeration, and analytical methods can be found in the Methods section and Supplementary Note 1.

As can be seen in Fig. 2A, cell numbers determined by the Most Probable Number (MPN, described in Methods section) for *Eucapsis* decreased by day 2 for all ratios of regolith:water by an order of magnitude from $\sim 10^5$ to $\sim 10^4$ cells mL$^{-1}$. *Eucapsis* growth from day 2 to day 22 fluctuated in all ratios between $10^3$ and $10^5$ cells mL$^{-1}$, with the most stable cell counts being observed in ratios 1:5 and 1:10 on days 2, 6, 10, and 18. This was in contrast to the *Chr20* (Fig. 2B) which, showed a small decrease in cell numbers on day 2 and only observed in the 2:1 ratio. Further, there is an increase in the MPN across the ratios by as much as an order of magnitude, suggesting that the regolith was beneficial to the extremophile, unlike in *Eucapsis* which slowly was declining. For *Chr20*, while there was an increase in cell numbers throughout the experiment, toward days 18 and 22 the lower ratios of 2:1 and 1:2 exhibited some fluctuations. Additionally, multiple ratios including the 2:1, statistically predicted MPN values approaching the maximum counted which would suggest that the presence of regolith was beneficial to the extremophile regardless of the water ratio.

Overall, for *Eucapsis*, there appeared to be a notable decline in the cell counts across all the ratios. In addition to *Eucapsis* not being an extremophile cyanobacterium, it is likely that the decrease of bioavailable nutrients contributed to a decline of growth over time. While *Eucapsis* declined with an average MPN of $\sim 10^4$ cells mL$^{-1}$, *Chr20* rapidly plateaued with an average MPN of $\sim 10^5$ cells mL$^{-1}$. This suggests that *Chr20* fared much better than *Eucapsis* by an order of magnitude and was able to survive and proliferate beyond its initial viable count regardless of the regolith:water ratio for the 22 days, while *Eucapsis* began to decline with a few exceptions.

Cyanobacterial control vials were also set up with either 20 mg of regolith or 1 mL of DI water-only and 5 μL of added inoculum

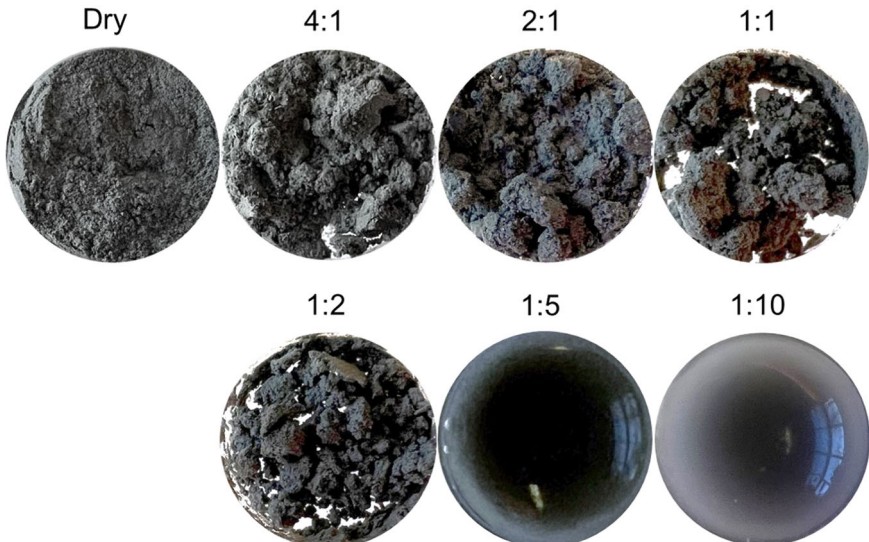

**Fig. 1 Appearance of regolith samples at different regolith:water ratios.** The 20 mg EETA79001 sawdust regolith samples were prepared in triplicate before homogenization by vortexing, dry, and with water added to give decreasing regolith:water (wt:vol) ratios of 4:1, 2:1, 1:1, 1:2, 1:5, and 1:10.

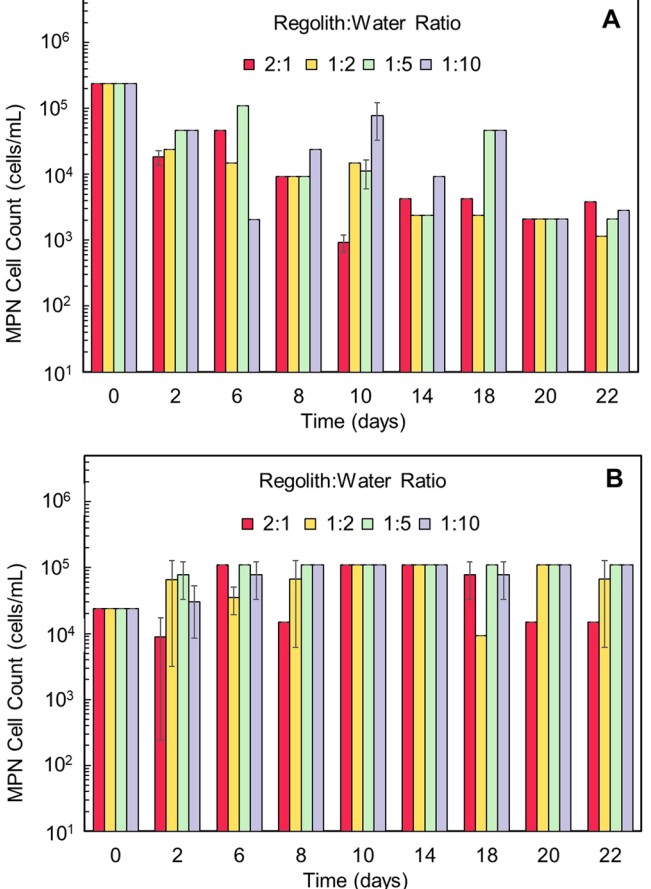

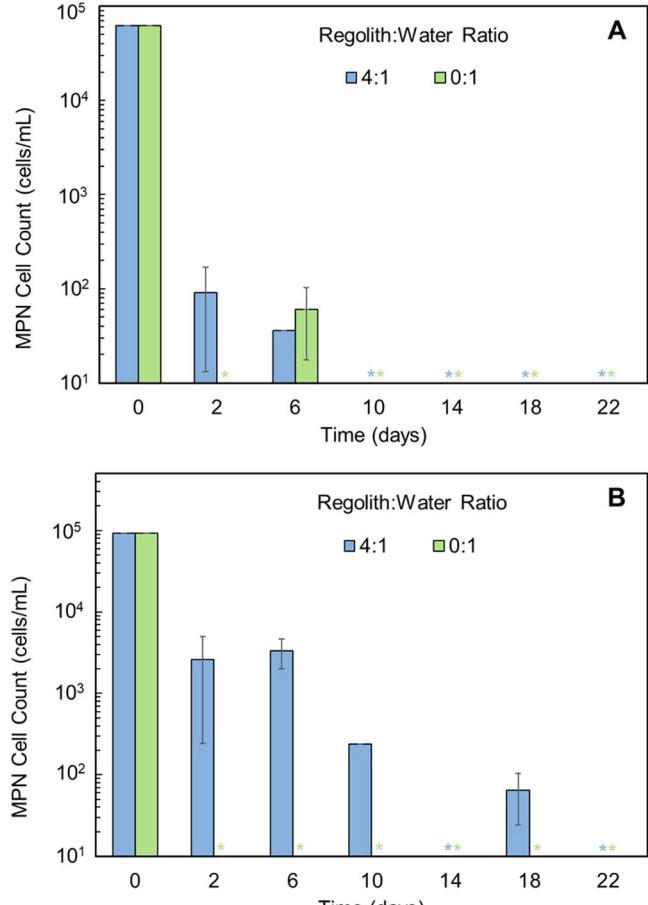

**Fig. 2 Growth of *Eucapsis* and *Chr20* over 22 days at different regolith:water ratios. A** *Eucapsis* at inoculation started at a cell count of ~2.5 × 10⁵ cells mL⁻¹ determined by the *Most Probable Number* (MPN) method. For all ratios, MPNs quantified decreased by two orders of magnitude between days 2 and 22. **B** *Chr20* at inoculation on day 0 started at a cell count of ~2.5 × 10⁴ cells mL⁻¹ determined by the MPN method with growth remaining constant between days 2 and 22, particularly for the 1:5 and 1:10 ratios. Each regolith:water ratio was performed for $n = 2$ and standard deviation was used for error bars.

**Fig. 3 Growth of *Eucapsis* and *Chr20* over 22 days at ratios that served as regolith and water controls. A** *Eucapsis* displayed minimal growth in both regolith-only and water-only, cell numbers dropped by three orders of magnitude for the regolith 4:1 control on days 2 and 6. Positive counts were collected for day 6 in the water control, however, beyond day 6 there was no detected growth. **B** *Chr20* showed a decrease in the regolith control by an order of magnitude until day 10. Growth was seen on day 18, but not to that seen earlier. In contrast, in the water control it showed no growth at any point. Overall, it exhibited higher counts and for longer periods than *Eucapsis* in regolith while neither demonstrated meaningful growth in the water control. Each regolith or water control test was performed for $n = 2$ and standard deviation was used for error bars. *Cell count was below the limits of quantification (LOQ).

solution (Fig. 3A, B). Neither *Eucapsis* nor *Chr20* showed growth in water alone—save for the *Eucapsis* day 6 timepoint. *Chr20* persisted with far higher MPN counts and for a longer period of time than the *Eucapsis* in the 4:1 regolith control, and although day 14 did not yield an MPN count for *Chr20*, growth was noted for day 18. This control demonstrated the longer persistence and survivability of the *Chr20* isolate than anticipated under limited bioavailable conditions and emphasizes its long-term durability. These controls suggest that the water itself could not sustain the cyanobacteria nor could the regolith provide an environment for meaningful growth. Instead, the growth at each ratio and timepoint of Fig. 2 is the result of the interaction between regolith and water fostering the conditions needed for viability and beneficial to *Chr20* as it surpassed the day 0 viable count for most ratios during the experimental duration. The somewhat fluctuating trend observed for both cyanobacteria suggests that perhaps either nutrients may have become depleted or, since cyanobacteria have a slower generation time compared to conventional bacteria (especially *Chr20*), a longer experimental duration would have provided further insight.

In comparison of the current findings with data obtained in previous work that evaluated bacterial growth in *Mars Mojavi Simulant* (MMS), *Mars Global Simulant* (MGS), and *NASA-JSC*

*Mars-1 Simulant* (JSC)[18], the *Chr20* in those studies (identified therein as *Gloeocapsa* prior to a whole genome analysis) maintained good growth over the 21 days in the MGS and JSC simulants for all regolith:water ratios, though less growth in the MMS simulant for the 2:1 regolith:water ratio. *Eucapsis* growth, however, dropped by several orders of magnitude by day 7 in the MMS and JSC and was not detectable in the MGS by several orders of magnitude. There was a slight increase in cell counts by day 21, but the cell recovery did not reach the initial day 0 levels. This implies that the martian regolith provided a better growth medium for both cyanobacteria, but that in the simulants, the extremophile *Chr20* fared much better, as might be expected. Both cyanobacteria showed a slight discernible drop in cell numbers between day 2 to day 22, with *Chr20* displaying good recovery between day 6 and day 14. As can be seen in Fig. 2, the regolith:water ratios showed little impact on the continued growth of the cyanobacteria following inoculation, suggesting that water plays a minimal role in their survivability.

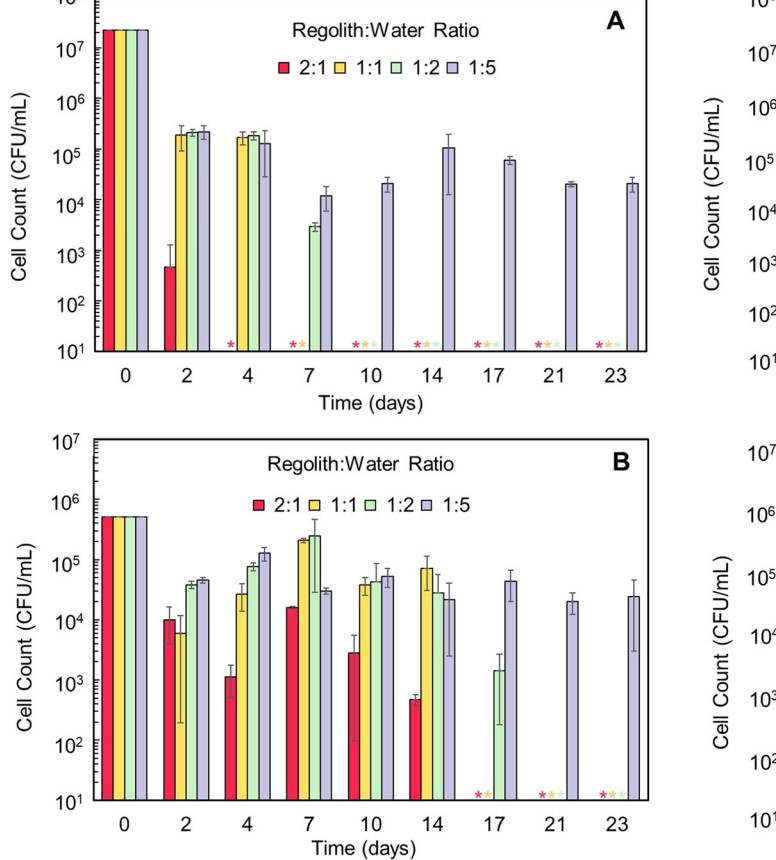

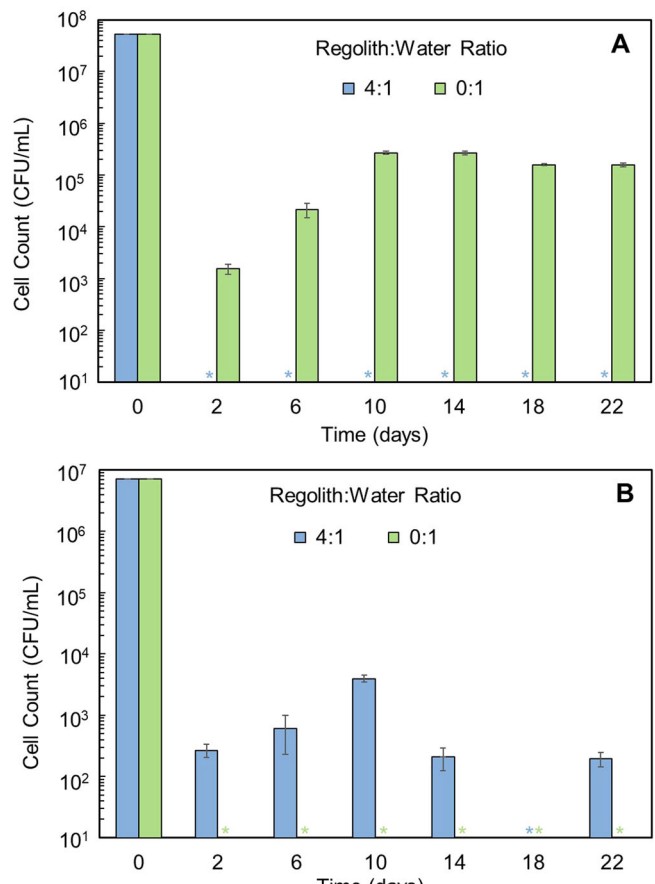

**Fig. 4 Growth of *E. coli* and *P. halocryophilus* over 23 days at different regolith:water ratios. A** *E. coli* growth decreased by day 4 for the 2:1 regolith:water ratio resulting in no cells being detected. After day 7 the 1:5 ratio continued to maintain consistent growth for the rest of the experiment. **B** *P. halocryophilus* displayed growth at all ratios until day 14 with the most significant growth being observed in the 1:1 and 1:2 regolith:water ratios and was most pronounced between days 7 and 14. The 1:5 ratio displayed a similar trend and by day 21 it was the only ratio still showing growth. Each regolith:water ratio was performed for $n = 3$ and standard deviation was used for error bars. *Cell count was below the limits of quantification (LOQ).

**Fig. 5 Growth of *E. coli* and *P. halocryophilus* over 22 days at ratios that served as regolith and water controls. A** *E. coli* in regolith control 4:1 ratio showed a significant decline from the viable count on day 0 with no growth detected at any point afterwards. *E. coli* in water control 0:1 displayed a decline of several orders of magnitude but plateaued in water by day 10 with growth maintained at ~$10^5$ CFU mL$^{-1}$. **B** *P. halocryophilus* in contrast did not show any growth in the 0:1 water control. In the 4:1 regolith control its cell numbers dropped significantly by day 2 but slowly increased by day 10. It also maintained limited though continuous growth in the regolith alone. The regolith directly impacts the continued growth of the bacteria as water alone is not enough to sustain it. Each regolith or water control test was performed for $n = 3$ and standard deviation was used for error bars. *Cell count was below the limits of quantification (LOQ).

**Bacteria: *E. coli* and *P. halocryophilus*.** The growth and survival of *E. coli* and *P. halocryophilus* in the EETA79001 regolith under terrestrial conditions were evaluated for up to 23 days at regolith:water ratios of 2:1, 1:1, 1:2, and 1:5. Similar to the cyanobacteria, the 4:1 ratio was designated as a dry regolith control containing only the bacterial inoculum and a 0:1 ratio was used as water control containing no regolith. A complete description of the culturing, growth, enumeration, and analytical methods, can be found in the Methods section and Supplementary Note 1. For *E. coli* there was a significant decrease in Colony-Forming Units (CFU) compared to the viable counts on day 0, similar to what was seen for the cyanobacteria's initial decline. The CFU mL$^{-1}$ count for *E. coli* in the regolith initially maintained sustained growth at regolith:water ratios of 1:1, 1:2, and 1:5 with less favorable cell counts for the 2:1 ratio (Fig. 4A). By day 4 the cell numbers for the 2:1 samples were below the limit of quantification (LOQ) and by day 10, only the 1:5 samples displayed growth, which steadily remained 1-2 orders of magnitude below the initial levels.

In contrast, as can be seen in Fig. 4B, it is apparent that the extremophile *P. halocryophilus* growth and survival was

significantly better in the regolith at all ratios than the *E. coli* and showed strong growth overall. Interestingly, even the 2:1 samples displayed significant growth between days 2–14 before decreasing below the LOQ. Throughout days 17-23, robust growth continued for the 1:5 sample. While the ratios of 2:1, 1:1, and 1:2 ultimately did not survive beyond days 14 and 17, their cell counts closely followed or exceeded those of the 1:5 sample.

Control vials were also set up to observe the effect of regolith or water alone on the bacteria. In Fig. 5A, *E. coli* displayed a decline from the viable count on day 0, reaching a plateau in the water-only sample by day 10 and then maintained its growth. While it is unsurprising that *E. coli* showed stable growth in water, for the regolith-only ratio of 4:1 there was not any quantified cell growth for the timepoints which suggests that *E. coli* survivability is more closely connected with the availability of water than martian regolith. In contrast, *P. halocryophilus* (Fig. 5B) exhibited no growth in the water controls but maintained limited but continued growth in the regolith-only samples. Comparing this data with the low regolith:water ratios of 2:1 and 1:1 in Fig. 4B

which only displayed growth up until day 14, the control data suggest that *P. halocryophilus* persisted to some extent well into day 22. By comparing the two, it is unmistakable that the regolith directly impacts the continued growth of the bacteria while water alone is not sufficient to sustain the extremophile and it is the regolith providing a growth substrate instead. Since extremophiles thrive in harsh environments the better performance of *P. halocryophilus* in regolith alone is not unexpected.

Similar to the previously performed experiments in martian simulants[18], where cell numbers initially increased significantly in most of the regolith:water ratios for all three simulants (MMS, MGS, and JSC) but by day 7, *E. coli* cells were only detected in the 1:5 sample for MMS. In JSC, viable cells were still detected in the 1:1, 1:2, and 1:5 samples at the day 7 and 14 timepoints. This was followed closely by MGS where although cell numbers had dropped by day 2, and then continued through days-7 and 14, only the 1:2 and 1:5 showed some recovery but was more obvious for the 1:5 ratio. It was also observed that *P. halocryophilus* showed some growth in the three simulants, and the growth was significantly better in JSC, followed by MGS, with MMS displaying the lowest growth. The previous results using Mars simulants and the current results for the EETA79001 regolith appear to show that *E. coli* is dependent on a higher regolith:water ratio for long-term survival regardless of the presence or absence of regolith.

The implication of these comparable results for *E. coli* and *P. halocryophilus* growth on martian regolith confirms our previous findings that *P. halocryophilus* has a greater survivability even at the lowest water ratios. This is most evident with the cell counts of 1:1 and 1:2, and to a lesser extent 2:1, between days 2 through 14 as these ratios fared far better than the counterpart samples with *E. coli*. Furthermore, the better growth of *P. halocryophilus* could be attributed due to its preference for a pH value of 7–8[50], while the optimum pH of *E. coli* is pH 5.6-6.5[51]. The EETA79001 regolith analysis showed that, in addition to the presence of ionic species that microorganisms could utilize, the regolith/water mixture with a pH 8 was closer to the optimum pH of *P. halocryophilus* and provided for better growth and survival. Such findings suggest that the extremophile's survivability was more related to the 20 mg of regolith than solely to the presence of water. The *E. coli* did not have cell counts above the LOQ for any ratios after day−10 except 1:5 while the same was not the case for *P. halocryophilus* until day 21. As the experimental vials were sealed with only parafilm during the experiment, it's possible that evaporation of water over the long duration timepoints may have caused an increase in the concentration of ionic species, which *P. halocryophilus* is better suited to handle[43,52]. The results show that a species with even better targeted adaptions could outperform *P. halocryophilus* and desert-dwelling cyanobacteria, and grow under stresses of moisture and salinity for longer durations and at higher cell counts in the martian regolith.

## Conclusions

The results of our study have implications for both microbial and human life on Mars. We have shown that martian regolith, in the form of EETA79001 sawdust, provides the necessary nutrients that support the growth of two cyanobacteria, *Chr20* and *Eucapsis*, and the extremophile bacterium *P. halocryophilus* for up to 22 and 23 days, respectively, at regolith:water ratios ranging from 4:1 to 1:10. Through previous work, we had identified the minimum growth requirements of these bacterial species[18] and had characterized the geochemical composition of EETA79001 leachate[33]. Building on that work, our current findings support the hypothesis that EETA79001-derived regolith not only

contains the necessary nutrients to support growth, but that they are bioavailable to bacteria, and that they may be further increased with the addition of water. More importantly, the EETA79001-derived martian regolith does not appear to contain any bactericidal substances, or at least in concentrations that could cause noticeable issues with cell growth. If the EETA79001 sawdust is broadly representative of the regolith on the martian surface, then our results imply that such life could have been present on Mars both, in the past during the warm and wet Noachian and Hesperian periods 3-4 billion years ago, and/or at present in appropriate subsurface aqueous environments.

The ability of cyanobacteria to grow in martian regolith also makes them ideal candidates for supporting human habitats on Mars. Recent research has indicated that cyanobacteria, functioning as an in-situ resource utilization (ISRU) system, could serve as the foundation for a biological life-support system by providing crucial nutrients for heterotrophic microbes[53]. Through this symbiotic relationship, they would not only sustain the growth of these heterotrophic microbes but also enable the production of plants as a vital source of food for humans on Mars. Furthermore, as a byproduct of their metabolic processes, cyanobacteria generate oxygen, thereby fulfilling another requirement for long-term sustainability of human habitats on Mars.

## Methods

**Mars meteorite sawdust and leachate.** The mineralogical composition of the EETA79001 Mars meteorite has been previously described in detail[28,54]. Briefly, it is composed of a primary basaltic host of medium-grained, feldspathic pyroxenite with olivine megacrysts and a minor component of pyroxene (lithology-A); a coarser-grained basalt similar to lithology-A but free of olivine megacrysts (lithology-B); and several shock-melted glass pockets and glass-filled veins (lithology-C). The detailed procedures and analysis of the leachate of the EETA79001 sawdust have been previously published[30,33]. The ionic species and their concentration in the leachate included those shown in Supplementary Table S1.

**Materials.** An autoclave (Harvey SterileMax, Marshall Scientific, NH, USA), incubator (Incufridge Professional Model 233, Revolutionary Science, Shafer, MN), and sterile Nanopure® 18.2 MΩ-cm deionized (DI) water, were used for all microbiological assays. Sterile Alga-Gro® media was used for all cyanobacterial assays (Carolina Biological Supply Company, Burlington, NC, USA). Tryptic Soy Agar (TSA) plates (Sigma Aldrich) were used for bacterial culture, maintenance, and experiments.

**Selection of microorganisms.** The rationale for the selection of the bacterial species and strains is described in Supplementary Note 1 and follows in parallel to the species used in Naz et al.[18,55]. (Note that the *Gloeocapsa-20201027-1* sp. isolate has now been identified more accurately via whole genome sequencing analysis and is referred to in this paper as *Chr20-20201027-1* (*Chr20*), with NCBI Accession Number: PRJNA868301 ID: 868301). The *Escherichia coli* B (*E. coli*) (#124300) and *Eucapsis* sp. (#151768) were purchased from Carolina Biological Supply Company (Burlington, NC, USA). The *P. halocryophilus* (*P. halocryophilus*) was generously donated by Dirk Schulze-Makuch (Technical University Berlin). The *Chr20* was isolated from a quartz rock collected in the Yungay Salar, located about 60 km inland in the hyperarid core of the Atacama Desert, Chile[55].

**Cyanobacterial culturing and enumeration.** During the experimental duration, axenic cultures of *Eucapsis* and *Chr20* cultures

were grown and maintained in 50 mL stocks in Alga-Gro media at room temperature (23 °C) in a foil-lined culture box fitted with a plant growth lamp ($\lambda = 455$–650 nm) and set to a cycle of 16-hour day light and 8-hour dark. For experiments, 4.5 mL of each cyanobacterium culture was equally distributed into three 1.5 mL conical microcentrifuge tubes and centrifuged using an Eppendorf benchtop 5424 R at 2000 rpm (448 g) (Fisher Scientific) for 5 minutes. The supernatant was discarded and replaced with 1 mL of DI water and centrifuged again. This wash step was repeated five times and the pellets were recombined and added to a single sterile microcentrifuge tube and resuspended in 1 mL of DI water. The suspension was mixed by pipetting or using a vortex mixer (VWR Mini Vortex, Radnor PA, USA). The optical density (OD) at a wavelength of 750 nm ($OD_{750}$) was determined by diluting in a 1:10 solution, 100 μL of cyanobacterium suspension with 900 μL of sterile DI water in a cuvette (Fisher Scientific, 14-955-127). The $OD_{750}$ was measured on an Agilent Cary 60 UV-Vis Spectrophotometer, calibrated with baseline subtraction to remove absorption of blank water samples, and cell numbers were adjusted to provide a cell count of ~$10^5$ to $10^6$ cells mL$^{-1}$ using the formula below to establish the day 0 viable count. The required inoculum was modified to produce the desired $OD_\lambda$ of 0.1 at 750 nm, calculated with the formula:

$$V_R = \frac{I_R \times V_F \times 1000}{10 \times OD_\lambda}$$

where $V_R$ is the volume required, $I_R$ is the inoculum required at $OD_\lambda$, $V_F$ is the final volume, and $OD_\lambda$ is the optical density at a wavelength of $\lambda$.

5 μL of the 0.1 $OD_{750}$ inoculum was added to the cyanobacteria and regolith in the 1 Dram (3.7 mL) vials (Fisher Scientific). The appropriate volume of DI water was then added to the 20 mg regolith samples in addition to the inoculum to give a final regolith:water (wt:vol) ratios of 4:1 (5 μL), 2:1 (10 μL), 1:2 (40 μL), 1:5 (100 μL), and 1:10 (200 μL). As no additional DI water was added to the 4:1 ratio beyond the required inoculum solution, this served as the control for regolith growth. In parallel, a water control 0:1, was prepared by adding 5 μL of inoculum to 1 mL of DI water. The appearance of the regolith in the vials at decreasing ratios of regolith:water is shown in Fig. 1.

Enumeration of *Eucapsis* and *Chr20* cells utilized the MPN method as previously described[56–59]. Briefly, it is a statistical technique used to estimate the concentration of microorganisms in a sample based on a series of dilutions prepared from the original sample and then inoculated into multiple replicate wells. The MPN method utilizes the concept of dilution to determine the MPN of microorganisms in the original sample. The presence or absence of microbial growth in the replicate wells is used to estimate the concentration of microorganisms in the original sample. The MPN value is derived from the number of positive, visibly growing, wells in each dilution. The more diluted the sample, the higher the dilution factor, and consequently, a lower number of positive results is expected. By considering the dilution factors and the number of positive results across multiple dilutions, the MPN can be calculated, providing an estimation of the concentration of microorganisms in the original sample. MPN is the method of choice to enumerate viable organisms in samples with low cell counts in water, particularly those containing particulate matter, which may prevent the ability to accurately count colonies.

The MPN technique was employed to measure viable cyanobacterial cells attached to the regolith particles. On the specified timepoint day, vials of regolith:water were resuspended in 1 mL of DI water, vigorously vortexed for 1 minute, and 100 μL of solution was serially diluted 1:10 into six wells of a 48-well plate containing 900 μL of Alga-Gro per well. Each dilution was

diluted by an order of magnitude with quadruple sets of technical replicates. Additionally, three wells were filled with 900 μL of Alga-Gro and 5 μL of prepared inoculum solution while three other wells contained only DI water. Such wells served as cyanobacterial positive and negative controls and validated the experimental finding. The loaded trays were then sealed with Parafilm (Cole-Palmer, Illinois, USA) to prevent evaporation and incubated at room temperature with alternating light and dark cycles. MPNs were determined by visually inspecting the wells for positive (green) or negative (colorless) growth, providing a statistical estimate of the viable cell count with each treatment repeated in quadruplicate. These values were derived from the equation specified in Appendix 2 of the U.S. Food and Drug Administration (FDA) Center for Food Safety and Applied Nutrition (CFSAN) Bacteriological Analytical Manual (BAM)[60]. The methods and steps of cyanobacterial resuspension, serial dilution in a well plate, and MPN enumeration are visualized in an infographic in Supplementary Figs. S1A and S1B.

**Bacterial culturing and enumeration.** *E. coli* and *P. halocryophilus* cultures were resuscitated from frozen stocks with both bacteria grown and maintained on Tryptic Soy Agar (TSA) plates (Sigma Aldrich). Prior to experimentation, a 20-hour culture of each bacteria was streaked onto a fresh TSA plate and incubated at 25 °C. At the 20-hour timepoint, the bacteria were collected with a sterile 10 μL loop (Thomas Scientific, Swedesboro, NJ) and suspended in 1 mL of DI water. The suspension was mixed either by pipetting or using a vortex mixer. The OD was measured with a spectrophotometer at 600 nm wavelength ($OD_{600}$) which was determined by diluting in a 1:10 solution, 100 μL of the bacterial suspension was added to 900 μL of DI water in a cuvette. The OD was calculated, as previously described, to produce a bacterial cell population of ~$10^7$ to $10^8$ cells mL$^{-1}$, prior to inoculation, viable counts were performed to provide the day 0 values.

The regolith samples were prepared by adding 20 mg of EETA79001 sawdust to each 1 Dram vial which was autoclaved 24 hours prior to use to remove any biological contaminants. 5 μL of the 0.1 $OD_{600}$ inoculum was pipetted into these vials then the appropriate volume of DI water was then added to the 20 mg regolith samples in addition to the inoculum to give regolith:water (wt:vol) ratios of 4:1 (5 μL), 1:1 (20 μL), 1:2 (40 μL), 1:5 (100 μL), for *E. coli* and *P. halocryophilus*. Each regolith:water ratio was prepared in triplicate for both bacteria. Figure 1 shows the regolith appearance with decreasing ratios of regolith:water. For the 4:1 ratio samples, no additional DI water was added to the vials besides the required inoculum solution, thus this served as the regolith control. The water control was prepared by adding 5 μL of inoculum to 1 mL of DI water and was denoted as 0:1 given its lack of regolith.

All inoculated Dram vials were prepared simultaneously on day 0, then on the day of a specified timepoint, 1 mL of DI water was mixed into the inoculated regolith to make a suspension. This stock suspension was then serially diluted 1:10 into microcentrifuge tubes preloaded with 900 μL of DI water three times, with the final solution being a thousandth of the undiluted concentration. From each microcentrifuge tube, 50 μL of the solution was cultured on conventional TSA agar plates and enumerated using CFU for both *E. coli* and *P. halocryophilus* cells.

The calculations for CFUs involves scaling up the number of colonies counted on a plate accounting for the dilution factor and the volume plated. The resulting CFU counts can then be scaled to represent the number of CFU present in the original bacterial suspension on a given timepoint. This scaling is performed to express the CFU count as a concentration per unit of volume (CFU mL$^{-1}$) which can then be quantitatively compared with

samples of different regolith:water ratios. The formula can be expressed as:

$$CFU\,mL^{-1} = N_C \div (D_F \times V_P)$$

where $N_C$ is the number of colonies counted, $D_F$ is the dilution factor, and $V_P$ is the volume plated. Selecting the agar plate with the best countable colonies was then adjusted to $CFU\,mL^{-1}$ to account for the differences in regolith:water ratios and compared directly with the day 0 viable counts. Each dilution was plated in triplicate sets and these replicates were then combined and averaged together for each ratio and timepoint day while the standard deviation was used for error bars. The methods and steps of bacterial resuspension, serial dilution, TSA plating, and CFU enumeration are visualized in an infographic in Supplementary Figs. S1C and S1D.

## Data availability

All reduced data are available in the main text and the Supplementary Information. All raw data will be permanently and publicly available on the Harvard Dataverse research data repository at https://dataverse.harvard.edu/dataverse/kounaves-lab.

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

## Acknowledgements
We thank Dr. Dongyu Liu for his valuable input at the start of this project and Professor Dirk Schulze-Makuch for generously providing the *P. halocryophilus*. Funding for parts of this research was provided by NASA Grant 80NSSC20K0460 through internal funding from Tufts University via indirect cost recovery return and the Laidlaw Foundation Scholars Program. The EETA79001 meteorite sawdust was obtained from the Astro-materials Curation Office at NASA-Johnson Space Center.

## Author contributions
N.N.: conceptualization, methodology, investigations, supervision, wrote original draft, and review/edited. B.H.: investigations, funding, and review/edited. J.N.: investigations, and review/edited. SPK: conceptualization, methodology, funding, supervision, wrote the original draft, and review/edited.

## Competing interests
The authors declare no competing interests.
