## [Peer Review File · Communications Earth & Environment]

6th Apr 23

Dear Professor Kounaves,

Please allow us to apologise for the delay in sending a decision on your manuscript titled "Microbial Growth in Actual Martian Regolith in the Form of Mars Meteorite EETA79001". It has now been seen by 3 reviewers, whose comments are appended below. You will see that they find your work of some potential interest. However, they have raised quite substantial concerns that must be addressed. In light of these comments, we cannot accept the manuscript for publication, but would be interested in considering a revised version that fully addresses these serious concerns.

In particular, we will need to see that the revised manuscript meets the following editorial thresholds:

- * Provide additional evidence for your conclusion that Martian regolith can support life in the form of control experiments which enable you test your hypothesis and gain robust results.
- * Consider and account for the role that high quantities of water played in keeping the bacteria alive, rather than the Martian regolith, incorporate this into your conclusions and clarify where cell numbers have actually decreased rather than grown.
- * Provide thorough details on the set-up and methodology of your incubation experiments, to the extent that they could be reproduced by readers.

We hope you will find the reviewers' comments useful as you decide how to proceed. Should additional work allow you to address these criticisms, we would be happy to look at a substantially revised manuscript. If you choose to take up this option, please either highlight all changes in the manuscript text file, or provide a list of the changes to the manuscript with your responses to the reviewers.

If the revision process takes significantly longer than three months, we will be happy to reconsider your paper at a later date, as long as nothing similar has been accepted for publication at Communications Earth & Environment or published elsewhere in the meantime.

We understand that due to the current global situation, the time required for revision may be longer than usual. We would appreciate it if you could keep us informed about an estimated timescale for resubmission, to facilitate our planning. Of course, if you are unable to estimate, we are happy to accommodate necessary extensions nevertheless.

Please use the following link to submit your revised manuscript, point-by-point response to the

reviewers' comments with a list of your changes to the manuscript text (which should be in a separate document to any cover letter) and any completed checklist:

[link redacted]

Please do not hesitate to contact me if you have any questions or would like to discuss the required revisions further. Thank you for the opportunity to review your work.

Best regards,

Huai Chen, PhD
Editorial Board Member
Communications Earth & Environment
orcid.org/0000-0001-7650-289X

Joe Aslin
Senior Editor
Communications Earth & Environment

EDITORIAL POLICIES AND FORMAT

If you decide to resubmit your paper, please ensure that your manuscript complies with our editorial policies and complete and upload the checklist below as a Related Manuscript file type with the revised article:

Editorial Policy Policy requirements (Download the link to your computer as a PDF.)

For your information, you can find some guidance regarding format requirements summarized on the following checklist:(<https://www.nature.com/documents/commsj-phys-style-formatting-checklist-article.pdf>) and formatting guide (<https://www.nature.com/documents/commsj-phys-style-formatting-guide-accept.pdf>).

REVIEWER COMMENTS:

Reviewer #1 (Remarks to the Author):

This paper presented the first use of actual martian regolith, in the form of Mars meteorite EETA79001 sawdust, to demonstrate its ability to support growth of four bacteria. In particular, the authors found the growth and survival of the four bacteria, *E. coli*, *Eucapsis* sp., Chr20-20201027-1, and *P. halocryophilus*, for up to 23 days under terrestrial conditions using regolith:water ratios from

4:1 to 1:10. Their results emphasized microbial life under appropriate conditions could have been present on Mars in the past and/or today and that the regolith does not contain any bactericidal agents. This is a very interesting research and offered a novel insight on building bio-sustainable human habitats on Mars.

The paper can be considered for publication in *Communications Earth & Environment* with the following points addressed.

1. I would suggest re-organizing the section of results and discussion, e.g., summarize and discuss the main points from this study rather than just discussing everything in order. Also, having subtitle for the discussion would be helpful so that you can guide the readers about which topic it is going to talk
2. Incubation. More details are needed to describe the set-up of the incubation experiment.

Reviewer #2 (Remarks to the Author):

The manuscript "Microbial Growth in Actual Martian Regolith in the Form of Mars Meteorite EETA79001" describe the possibility of life on Mars by inoculation bacterial strains. It is a quite interesting topic for public. Yet, there are some major concerns for bacterial growth.

- 1) The methods part is quite confused. From my understanding, author inoculated 5 ul 1:10 bacterial solution into the mars sawdust. E.coli can only grow in 1:5 Soil: water medium, it is hard to discern the increases in the E.coli number were caused by water or mars sawdust. E.coli are easy to grow in water. Some control experiments, such as only growing in water, are needed.
- 2) Compared to day 0, the number of Eucapsis and Chr20 decreased across all incubation period. With no increasing number of cells, it is lack of evidence to support the hypothesis that martian regolith can support bacterial growth. Maybe you use some black yeast which they can grow in the earth rock to test this hypothesis.

Reviewer #3 (Remarks to the Author):

Neveda et al., used actual Martian regolith to grow living materials. It is an interesting work and several bacteria did survive on the Martian medium. The results indicated that human beings might be able to grow Earth's lives on Mars while building bio-sustainable human habitats. However, there are still some big concerns for the bacterial growth in this study. 1) The start inoculation of Eucapsis and Chr20 is ca. 1000 cells, but remained ca. 100 cells in only two days. The method for cell counting looks insufficient to harvest most bacteria in the medium. Anyway, no increasing of cell numbers of extremophiles did not support the growth of bacteria. 2) The long-term cultivation of E. coli and P. holi needs extreme high water: soil ratio. It seems water is more suitable to grow bacteria rather than using Martian soil. It is hard to determine whether Martian soil or water that supports the long-time survival of these two bacteria. Still, the results did not provide sufficient evidence that these four bacteria can be long-term cultivated on Mars. I have several other major suggestions as following:

- 1) To detect the survival of long-generation extremophiles, you might want to use acridine orange as indicator for living cells.

2) Some rock-inhabited species, such as Dothideomycetes fungi or lichen might get cultivation on the extreme nutrient-poor Martian materials.

3) Some control experiments, e.g. using Earth rocks, sterilized water, and synthetic liquid medium with Martian salt composition, will drive more convincing conclusions.

Responses to Reviewers' Comments - Manuscript COMMSENV-23-0167-T

We are sincerely thankful to the reviewers for their time and thoughtful feedback, which has helped to strengthen our manuscript. We have made significant changes to the manuscript and have fully addressed all of the reviewers' comments, documented below in blue. All changes to the text have also been provided in a Word track-edited manuscript file.

Reviewer #1:

This paper presented the first use of actual martian regolith, in the form of Mars meteorite EETA79001 sawdust, to demonstrate its ability to support growth of four bacteria. In particular, the authors found the growth and survival of the four bacteria, *E. coli*, *Eucapsis* sp., *Chr20-20201027-1*, and *P. halocryophilus*, for up to 23 days under terrestrial conditions using regolith:water ratios from 4:1 to 1:10. Their results emphasized microbial life under appropriate conditions could have been present on Mars in the past and/or today and that the regolith does not contain any bactericidal agents. This is very interesting research and offered a novel insight on building bio-sustainable human habitats on Mars. The paper can be considered for publication in Communications Earth & Environment with the following points addressed.

1. I would suggest re-organizing the section of results and discussion, e.g., summarize and discuss the main points from this study rather than just discussing everything in order. Also, having subtitle for the discussion would be helpful so that you can guide the readers about which topic it is going to talk

We have reorganized the Results & Discussion section and grouped findings into two main themes “*E. coli* and *P. halocryophilus*” and “Cyanobacteria: *Eucapsis* and *Chr20*” and added subtitles to delineate these findings.

2. Incubation. More details are needed to describe the set-up of the incubation experiment.

Done. We have added additional information to the methods section to describe our incubation procedures in greater detail, and have also designed new figures illustrating the procedures, which have been added to the Supplementary Information.

Reviewer #2:

The manuscript “Microbial Growth in Actual Martian Regolith in the Form of Mars Meteorite EETA79001” describe the possibility of life on Mars by inoculation bacterial strains. It is a quite interesting topic for public. Yet, there are some major concerns for bacterial growth.

1) The methods part is quite confused. From my understanding, author inoculated 5 ul 1:10 bacterial solution into the mars sawdust. E.coli can only grow in 1:5 Soil: water medium, it is hard to discern the increases in the E.coli number were caused by water or mars sawdust. E.coli are easy to grow in water. Some control experiments, such as only growing in water, are needed.

We agree, thus we have added more information to the methods section to describe our incubation procedures in greater detail, and have added new figures illustrating the procedures to the Supplementary Information. We have also conducted an additional set of control experiments as suggested, including experiments to observe growth and survival in water-only. As pointed out by this reviewer, *E. coli* was able to survive and grow in the water-only samples, although the other three bacterial strains (*P. halocryophilis*, *Eucapsis*, and *Chr20*) did not survive.

2) Compared to day-0, the number of *Eucapsis* and *Chr20* decreased across all incubation period. With no increasing number of cells, it is lack of evidence to support the hypothesis that martian regolith can support bacterial growth. Maybe you use some black yeast which they can grow in the earth rock to test this hypothesis.

Thank you for the suggestion to assess the growth of black yeast. However, in the current work the scope by necessity was limited to bacterial species. Later investigations are planned for growth of higher order organisms such as black yeast.

There was an error on our part in using the somewhat complicated MPN software. This error has now been corrected and doublechecked. After reanalyzing the data, we have indeed found an increase in the MPN compared to baseline values, which does indicate growth. This has now been corrected in the text. While *Eucapsis* cells did reduce by day 23, MPN counts stabilized and saw limited fluctuation after day 10. Compared to the results of our new control experiments, particularly in water-only, where *Eucapsis* exhibited no survival past day 6, our original findings also support the hypothesis that the martian regolith can support growth. Similarly, in the water-only control experiments, *Chr20* displayed no survival past day 0, but stable and consistent survival through day 23 in regolith and water, particularly in higher ratios (1:5 and 1:10). These findings suggest the stability in both standard and extremophile cyanobacteria samples. Given that some cyanobacterial spp. are slower growing, longer duration experiments which were beyond the time constraints of this project may be needed to better elucidate the ability of martian regolith to support long term growth.

Reviewer #3:

Neveda et al., used actual Martian regolith to grow living materials. It is an interesting work and several bacteria did survive on the Martian medium. The results indicated that human beings might be able to grow Earth's lives on Mars while building bio-sustainable human habitats. However, there are still some big concerns for the bacterial growth in this study.

1) The start inoculation of *Eucapsis* and *Chr20* is ca. 1000 cells, but remained ca. 100 cells in only two days. The method for cell counting looks insufficient to harvest most bacteria in the medium. Anyway, no increasing of cell numbers of extremophiles did not support the growth of bacteria.

There was an error on our part in using the somewhat complicated MPN software. This error has now been corrected and doublechecked. After reanalyzing the data, we have indeed found an increase in the MPN compared to baseline values, which does indicate growth. This has now been corrected in the text.

2) The long-term cultivation of *E. coli* and *P. halo* needs extreme high water:soil ratio. It seems water is more suitable to grow bacteria rather than using martian soil. It is hard to determine whether martian soil or water that supports the long-time survival of these two bacteria. Still, the results did not provide sufficient evidence that these four bacteria can be long-term cultivated on Mars.

We have clarified our results through additional control experiments using sterilized water alone, which we believe supports our original findings. While *E. coli* exhibited survival and growth with sterilized water alone, all other bacteria ultimately exhibited higher growth and longer survival in regolith and water combined. This indicates that nutrients present in regolith can support survival above and beyond what would be observed in sterilized water alone. Further, these results support the idea that, while minimum necessary nutrients may be present in regolith, water promotes the bioavailability of these nutrients over time, which can be explained by the higher water ratios exhibiting higher MPN counts.

Other suggestions:

1) To detect the survival of long-generation extremophiles, you might want to use acridine orange as indicator for living cells.

We are unable to incorporate acridine dye into this current work, although this is a very valuable consideration which we will consider in future research. There are several limitations that would need to be addressed to use acridine staining, given the nature of these experiments. The use of acridine dye would require either epifluorescent microscopy or flow cytometry, both of which can pose limitation to accurate enumeration when working with environmental samples containing soil particles (Frossard, et al., *Front. Microbio*, 7, 2016). While epifluorescent microscopy has been shown to be labor intensive and sensitive to observer bias, and may result in an overcount of cells, flow cytometry requires separation and filtering of regolith particles from the sample and methods to do this are still being

optimized. Any attempts to separate particulate and cellular matter may inadvertently lead to disposal of cells attached to regolith and under report the number of cells/colonies. In either case, incorporating dye into the enumeration process may result in false positives, if regolith particles included in the samples are mistakenly identified as cells. There are other potential limitations to using dye methods, which may yield ambiguous results if cell injury or chloroplasts of dead cells that still emit fluorescence interfere with live dead differentiation. Further, the presence of aggregates can result in inaccurate cells counts, and due the gelatinous nature of *Chr20* we expect that despite sonication and cell mixing there would inevitably be aggregates still present that affected the stain uptake and resulted in a lower or higher signal. Problems can also occur due to sonication, as excess exposure can lead to membrane permeabilization as well as the issue that older cells that display thicker membranes could lead to slower uptake of cell membrane dye.

- 2) Some rock-inhabited species, such as Dothideomycetes fungi or lichen might get cultivation on the extreme nutrient-poor Martian materials.

This is a fascinating idea which we hope to explore through future work. Through this current project, we aimed to explore survival and growth of bacterial species, only, and selected a range of bacteria which we hypothesized could grow easily in regolith due to their extremophile nature (e.g. *P. halocryophilis* and *Chr20*) as well as bacteria which are not known extremophiles (e.g., *Eucapsis* and *E. coli*) which are well known to require basic nutrients to survive, to see whether martian regolith could support these diverse requirements. We are enthusiastic about incorporating fungal species in future work as the next stage of this research.

- 3) Some control experiments, e.g. using Earth rocks, sterilized water, and synthetic liquid medium with Martian salt composition, will drive more convincing conclusions.

We designed a new set of control experiments using sterilized water only, which we have incorporated into the manuscript. Additionally, through previous published work (*N. Naz, et al., Microbial Growth in Martian Soil Simulants Under Terrestrial Conditions: Guiding the Search for Life on Mars. Astrobiology, 2022,1210-1221*) we have conducted several other relevant experiments. In this previous work we; 1) identified and characterized the minimum growth requirements of the two bacteria (*Escherichia coli* B and *Eucapsis* sp) and two extremophiles (*Gloeocapsa-20201027-1* sp and *Planococcus halocryophilus*) in a variety of growth media, and 2) studied growth and survival in three different synthetic Martian simulants, and 3) analyzed the geochemistry of leachates from the martian simulants. Our current research builds upon the findings of this prior work, and we have added a sentence to the conclusion to acknowledge this.

20th Sep 23

Dear Professor Kounaves,

Your manuscript titled "Microbial Growth in Actual Martian Regolith in the Form of Mars Meteorite EETA79001" has now been seen by our reviewers, whose comments appear below. In light of their advice we are delighted to say that we are happy, in principle, to publish a suitably revised version in Communications Earth & Environment under the open access CC BY license (Creative Commons Attribution v4.0 International License).

We therefore invite you to edit your manuscript to comply with our format requirements and to maximise the accessibility and therefore the impact of your work.

EDITORIAL REQUESTS:

*****Please take care to match our formatting and policy requirements. We will check revised manuscript and return manuscripts that do not comply. Such requests will lead to delays. *****

SUBMISSION INFORMATION:

OPEN ACCESS:

Communications Earth & Environment is a fully open access journal. Articles are made freely accessible on publication under a [CC BY license](http://creativecommons.org/licenses/by/4.0) (Creative Commons Attribution 4.0 International License). This license allows maximum dissemination and re-use of open access materials and is preferred by many research funding bodies.

For further information about article processing charges, open access funding, and advice and support from Nature Research, please visit <https://www.nature.com/commsenv/article-processing-charges>

At acceptance, you will be provided with instructions for completing this CC BY license on behalf of all authors. This grants us the necessary permissions to publish your paper. Additionally, you will be

asked to declare that all required third party permissions have been obtained, and to provide billing information in order to pay the article-processing charge (APC).

[link redacted]

Best regards,

Huai Chen
Editorial Board Member
Communications Earth & Environment

Joe Aslin
Senior Editor,
Communications Earth & Environment
<https://www.nature.com/commsenv/>
Twitter: @CommsEarth

REVIEWERS' COMMENTS:

Reviewer #1 (Remarks to the Author):

Consent to publication. This is a very interesting research and offered a novel insight on building bio-sustainable human habitats on Mars.

Reviewer #2 (Remarks to the Author):

I am satisfied with this version now.

Reviewer #3 (Remarks to the Author):

I want to thank the author for their detailed responses to all my requests. They managed to clearly address all my concerns. I am very happy with this response. I think this will be a great addition to the field and should be published without delay.